# Computing Valid $p$-value for Optimal Changepoint by Selective Inference using Dynamic Programming

**Vo Nguyen Le Duy**[*]
Nagoya Institute of Technology and RIKEN
duy.mllab.nit@gmail.com

**Hiroki Toda**[*]
Nagoya Institute of Technology
toda.h.mllab.nit@gmail.com

**Ryota Sugiyama**
Nagoya Institute of Technology
sugiyama.r.mllab.nit@gmail.com

**Ichiro Takeuchi**[†]
Nagoya Institute of Technology and RIKEN
takeuchi.ichiro@nitech.ac.jp

## Abstract

Although there is a vast body of literature related to methods for detecting change-points (CPs), less attention has been paid to assessing the statistical reliability of the detected CPs. In this paper, we introduce a novel method to perform statistical inference on the significance of the CPs, estimated by a Dynamic Programming (DP)-based optimal CP detection algorithm. Our main idea is to employ a Selective Inference (SI) approach — a new statistical inference framework that has recently received a lot of attention — to compute exact (non-asymptotic) valid $p$-values for the detected optimal CPs. Although it is well-known that SI has low statistical power because of over-conditioning, we address this drawback by introducing a novel method called parametric DP, which enables SI to be conducted with the minimum amount of conditioning, leading to high statistical power. We conduct experiments on both synthetic and real-world datasets, through which we offer evidence that our proposed method is more powerful than existing methods, has decent performance in terms of computational efficiency, and provides good results in many practical applications.

## 1 Introduction

Changepoint (CP) detection is a fundamental problem and has been studied in many areas. The goal of CP detection is to find changes in the underlying mechanism of the observed sequential data. Analyzing the detected CPs benefits to several applications [14, 35, 15, 22, 19]. There is a vast body of literature related to methods for detecting CPs [2, 47, 32, 42, 29, 11, 46] — nice surveys can be found in [1, 44]. CP detection is usually formulated as the problem of minimizing the cost over segmentations, where *Dynamic Programming (DP)* is commonly used because it can solve the minimization problem efficiently, and exactly find the optimal CPs under the given criteria.

Unfortunately, less attention has been paid to the statistical reliability of the detected CPs. Without statistical reliability, the results may contain many *false detections*, i.e., the detected CPs may not be true CPs. These falsely detected CPs are harmful when they are used for high-stake decision making such as medical diagnosis or automatic driving. Therefore, it is highly necessary to develop a *valid statistical inference* for the detected CPs that can properly control the risk of false detection.

Valid statistical inference on CPs is intrinsically difficult because the observed data is used twice — one for detection and another for inference, which is often referred to as *double dipping* [23]. In

---

[*]Equal contribution
[†]Corresponding author

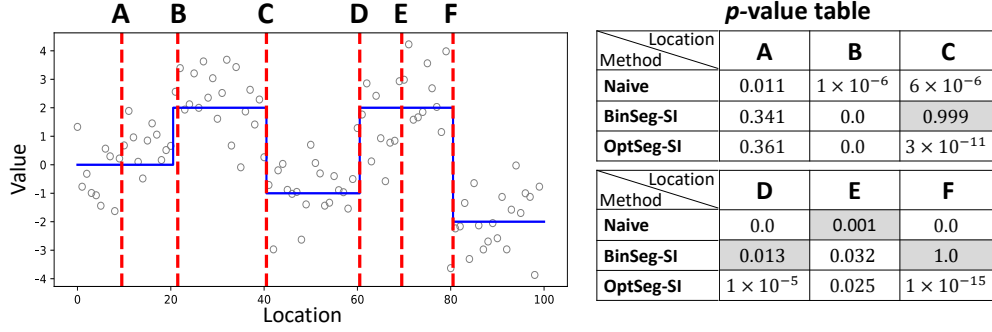

Figure 1: An illustrative example of the problem and the methods considered in this paper. The blue line and the grey circles indicate the underlying mean and the observed sequence, respectively. The red dotted lines are the results of optimal segmentation (OptSeg) and binary segmentation (BinSeg). Here, the results of OptSeg and BinSeg were the same. With Bonferroni correction, to control false detection rate at $0.05$, the significance level is decided by $\frac{0.05}{6} \approx 0.008$. The naive $p$-value is small even for falsely detected CP (**E**). BinSeg $p$-values can identify falsely detected CPs, but it fails to detect some true CPs (**C**, **D**, **F**) due to the lack of power. The proposed $p$-values (OptSeg-SI) can successfully identify both true positive and false positive detections.

statistics, it has been recognized that naively computing $p$-values in double dipping is highly biased, and correcting this bias is challenging. Our idea is to introduce *Selective Inference (SI)* framework for resolving this challenge.

**Existing works and their drawbacks.** In the case of testing for single CP, most of the existing inference methods rely on *asymptotic* distribution of the maximum discrepancy measure, such as CUSUM score [33], Fisher discriminant score [30, 16], and MMD [25], which is derived under some restrictive assumptions such as weak dependence among the data points. Asymptotic inference for multiple CPs was proposed by [14] under the name of *SMUCE*. These asymptotic approaches often fail to control type I error when the sequence is short or contains highly correlated data points. Besides, it has been observed that these approaches are often conservative, i.e., low statistical power [17].

In the past few years, SI has been actively studied and applied to various problems [3, 13, 8, 41, 7, 4, 28, 27, 34, 43, 24, 48, 38, 40, 9, 10]. The basic idea of SI is to make inference conditional on the selection event, which allows us to derive the *exact* (non-asymptotic) sampling distribution of test statistic. However, characterizing the necessary and sufficient selection event is computationally challenging. For example, in [24], the authors considered inference conditional not only on the selected features but also on their signs for computational tractability. However, such an *over-conditioning* leads to loss of power [24, 26, 9].

SI was first discussed in the context of CP detection problem by Hyun et al. [17], in which the authors studied Fused Lasso. Later, Umezu et al. [45] and Hyun et al. [18] studied SI for CUSUM-based CP detection and binary segmentation, respectively. Unfortunately, these methods inherit the drawback of other SI studies, i.e., the loss of power by over-conditioning. In other words, the inference is made not only conditional on the detected CPs, but also on other unnecessary extra events.

**Contributions.** We provide an exact (non-asymptotic) inference method for optimal CPs, which we call *OptSeg-SI*, based on the concept of SI. To our knowledge, this is the first method that can provide valid $p$-values to the CPs detected by DP. Unlike existing SI approaches for CPs [17, 18, 45], the inference in the OptSeg-SI method is made under the minimum amount of conditioning, leading to high statistical power. To this end, we develop a new method called *parametric DP*, which enables us to efficiently characterize the selection event. We conduct experiments on both synthetic and real-world datasets, by which, we offer the evidence that the OptSeg-SI 1) is more powerful than the existing methods [18, 14], 2) successfully controls false detection probability, 3) has good performance in terms of computational efficiency, and 4) provides good results in many practical applications.

Figure 1 shows an illustrative example of the problem and the methods we consider in this paper. For reproducibility, our implementation is available at

`https://github.com/vonguyenleduy/parametric_selective_inference_changepoint`

## 2 Problem Statement

We consider CP detection problem for mean-shift, which is the most studied model in the literature, and has been applied to many real-world applications, especially in bioinformatics [31, 6]. Mean-shift CP detection is the base of many other CP detection methods. If one knows what kind of changes to focus on (e.g., changes in variance), we can convert the problem into mean-shift CP detection. Otherwise, nonparametric CP detection methods such as kernel CP detection [25] can be used. It is well known that many nonparametric methods can be cast into a mean-shift CP detection. Therefore, mean-shift CP detection is worth investigating as a canonical form of the more complex problems.

Let us consider a random sequence $\boldsymbol{X} = (X_1, \ldots, X_N)^\top \sim \mathbb{N}(\boldsymbol{\mu}, \boldsymbol{\Sigma})$, where $N$ is the length, $\boldsymbol{\mu} \in \mathbb{R}^N$ is unknown mean vector, and $\boldsymbol{\Sigma} \in \mathbb{R}^{N \times N}$ is covariance matrix which is known or estimable from external data [3]. When there are multiple *true* CPs, the mean vector $\boldsymbol{\mu} = (\mu_1, \ldots, \mu_N)^\top$ is represented as a piecewise constant sequence having shifts at these CPs.

Given an observed sequence $\boldsymbol{x}^{\text{obs}} = (x_1^{\text{obs}}, \ldots, x_N^{\text{obs}})^\top \in \mathbb{R}^N$, the goal of CP detection is to estimate the true CPs. The vector of detected CP locations is denoted as $\boldsymbol{\tau}^{\text{det}} = (\tau_1^{\text{det}}, \ldots, \tau_K^{\text{det}})$, where $K$ is the number of CPs, and $\tau_1^{\text{det}} < \cdots < \tau_K^{\text{det}}$ are the CP locations (we set $\tau_0^{\text{det}} = 0$ and $\tau_{K+1}^{\text{det}} = N$). We define $\boldsymbol{x}_{s:e} \sqsubseteq \boldsymbol{x}$ as a subsequence of $\boldsymbol{x} \in \mathbb{R}^N$ from positions $s$ to $e$, where $1 \le s \le e \le N$. The average of $\boldsymbol{x}_{s:e}$ is written as $\bar{x}_{s:e} = \frac{1}{e-s+1} \sum_{i=s}^e x_i$, and the cost function which measures the "homogeneity" of $\boldsymbol{x}_{s:e}$ is defined as $C(\boldsymbol{x}_{s:e}) = \sum_{i=s}^e (x_i - \bar{x}_{s:e})^2$.

### 2.1 Optimal CP detection

When the number of CPs $K$ is fixed, the maximum likelihood estimation of the CPs is formulated as

$$\boldsymbol{\tau}^{\text{det}} = \arg \min_{\boldsymbol{\tau}} \sum_{k=1}^{K+1} C(\boldsymbol{x}_{\tau_{k-1}+1:\tau_k}^{\text{obs}}). \tag{1}$$

When the number of CPs $K$ is unknown, the penalized maximum likelihood estimation is defined as

$$\boldsymbol{\tau}^{\text{det}} = \arg \min_{\boldsymbol{\tau}} \sum_{k=1}^{\dim(\boldsymbol{\tau})+1} C(\boldsymbol{x}_{\tau_{k-1}+1:\tau_k}^{\text{obs}}) + \beta \dim(\boldsymbol{\tau}), \tag{2}$$

where $\dim(\boldsymbol{\tau})$ is the dimension of a CP vector $\boldsymbol{\tau}$, and $\beta \in \mathbb{R}^+$ is a hyper-parameter, which can be defined based on several methods such as BIC [36]. The optimal solutions of (1) and (2) can be obtained by DP.

**Definition 1.** *We denote the event that the optimal CP vector $\boldsymbol{\tau}^{\text{det}}$ is detected by applying DP algorithm $\mathcal{A}$ to the observed sequence $\boldsymbol{x}^{\text{obs}}$ as*

$$\boldsymbol{\tau}^{\text{det}} = \mathcal{A}(\boldsymbol{x}^{\text{obs}}). \tag{3}$$

### 2.2 Inference on the detected CPs

For the inference on the $k^{\text{th}}$ detected CP $\tau_k^{\text{det}}$, $k \in [K]$, we consider the following statistical test

$$\text{H}_{0,k} : \mu_{\tau_{k-1}^{\text{det}}+1} = \cdots = \mu_{\tau_k^{\text{det}}} = \mu_{\tau_k^{\text{det}}+1} = \cdots = \mu_{\tau_{k+1}^{\text{det}}}$$
$$\text{vs.} \tag{4}$$
$$\text{H}_{1,k} : \mu_{\tau_{k-1}^{\text{det}}+1} = \cdots = \mu_{\tau_k^{\text{det}}} \ne \mu_{\tau_k^{\text{det}}+1} = \cdots = \mu_{\tau_{k+1}^{\text{det}}},$$

where $[K] = \{1, ..., K\}$ indicates the set of natural numbers up to $K$. A natural choice of the test statistic is the difference between the average of the two segments before and after the $k^{\text{th}}$ CP

$$\boldsymbol{\eta}_k^\top \boldsymbol{X} = \bar{X}_{\tau_{k-1}^{\text{det}}+1:\tau_k^{\text{det}}} - \bar{X}_{\tau_k^{\text{det}}+1:\tau_{k+1}^{\text{det}}}, \tag{5}$$

where $\boldsymbol{\eta}_k = \frac{1}{\tau_k^{\mathrm{det}} - \tau_{k-1}^{\mathrm{det}}} \mathbf{1}_{\tau_{k-1}^{\mathrm{det}}+1:\tau_k^{\mathrm{det}}}^N - \frac{1}{\tau_{k+1}^{\mathrm{det}} - \tau_k^{\mathrm{det}}} \mathbf{1}_{\tau_k^{\mathrm{det}}+1:\tau_{k+1}^{\mathrm{det}}}^N$, and $\mathbf{1}_{s:e}^N \in \mathbb{R}^N$ is a vector whose elements from position $s$ to $e$ are set to 1, and 0 otherwise. Suppose, for now, that the hypotheses in (4) are fixed, i.e., non-random. Then, the *naive* (two-sided) $p$-value is given as

$$p_k^{\mathrm{naive}} = \mathbb{P}_{\mathrm{H}_{0,k}} \left( |\boldsymbol{\eta}_k^\top \boldsymbol{X}| \geq |\boldsymbol{\eta}_k^\top \boldsymbol{x}^{\mathrm{obs}}| \right) = 2 \min\{F_{0, \boldsymbol{\eta}_k^\top \Sigma \boldsymbol{\eta}_k}(\boldsymbol{\eta}_k^\top \boldsymbol{x}^{\mathrm{obs}}), 1 - F_{0, \boldsymbol{\eta}_k^\top \Sigma \boldsymbol{\eta}_k}(\boldsymbol{\eta}_k^\top \boldsymbol{x}^{\mathrm{obs}})\}, \quad (6)$$

where $F_{m,s^2}$ is the c.d.f. of Normal distribution $\mathbb{N}(m, s^2)$.

However, since the hypotheses in (4) are actually not fixed in advance, the naive $p$-value is not *valid* in the sense that, if we reject $\mathrm{H}_{0,k}$ with a significance level $\alpha$ (e.g., $\alpha = 0.05$), the false detection rate (type-I error) cannot be controlled at level $\alpha$. This is due to the fact that the hypotheses in (4) are *selected* by data, and *selection bias* exists. One way to avoid the selection bias is to consider the sampling distribution of a test statistic *conditional* on the selection event. Thus, we employ the following *conditional* $p$-value

$$p_k^{\mathrm{selective}} = \mathbb{P}_{\mathrm{H}_{0,k}} \left( |\boldsymbol{\eta}_k^\top \boldsymbol{X}| \geq |\boldsymbol{\eta}_k^\top \boldsymbol{x}^{\mathrm{obs}}| \mid \mathcal{A}(\boldsymbol{X}) = \mathcal{A}(\boldsymbol{x}^{\mathrm{obs}}), \boldsymbol{q}(\boldsymbol{X}) = \boldsymbol{q}(\boldsymbol{x}^{\mathrm{obs}}) \right), \quad (7)$$

where $\mathcal{A}(\boldsymbol{X}) = \mathcal{A}(\boldsymbol{x}^{\mathrm{obs}})$ indicates the event that the detected CP vector for a random sequence $\boldsymbol{X}$ is the same as the detected CP vector for the observed sequence $\boldsymbol{x}^{\mathrm{obs}}$. The second condition $\boldsymbol{q}(\boldsymbol{X}) = \boldsymbol{q}(\boldsymbol{x}^{\mathrm{obs}})$ indicates that the component which is independent of the test statistic $\boldsymbol{\eta}_k^\top \boldsymbol{X}$ for a random sequence $\boldsymbol{X}$ is the same as the one for $\boldsymbol{x}^{\mathrm{obs}}$ [4]. The $\boldsymbol{q}(\boldsymbol{X})$ corresponds to the component $\boldsymbol{z}$ in the seminal paper (see [24], Sec 5, Eq 5.2 and Theorem 5.2), and it is given by

$$\boldsymbol{q}(\boldsymbol{X}) = (I_N - \boldsymbol{c}\boldsymbol{\eta}_k^\top)\boldsymbol{X} \ \text{ where } \boldsymbol{c} = \Sigma \boldsymbol{\eta}_k (\boldsymbol{\eta}_k^\top \Sigma \boldsymbol{\eta}_k)^{-1}.$$

The $p$-value in (7) is called *selective type I error* or *selective p-values* in SI literature [12]. Figures 8 and 9 in Appendix A.4 show the distribution of naive $p$-values and selective $p$-values when the null hypothesis $\mathrm{H}_{0,k}$ is true. The naive $p$-values are not uniformly distributed, while selective $p$-values are. The uniformly distributed property is necessary for valid $p$-values since it indicates

$$\mathbb{P}_{\mathrm{H}_{0,k}} \left( p_k^{\mathrm{selective}} < \alpha \right) = \alpha, \quad \forall \alpha \in [0, 1].$$

Our contribution is to provide an efficient method for computing selective $p$-value in (7) by characterizing the selection event $\mathcal{A}(\boldsymbol{X}) = \mathcal{A}(\boldsymbol{x}^{\mathrm{obs}})$, which is computationally challenging because we have to find the whole set of sequences in $\mathbb{R}^N$ having the same optimal CP vectors on $\boldsymbol{x}^{\mathrm{obs}}$.

## 3 Proposed Method

We propose a method for computing selective $p$-values in (7). We focus here on the case where the number of CPs $K$ is fixed. The case for unknown $K$ will be discussed in §4. Figure 2 shows the schematic illustration of the OptSeg-SI method.

### 3.1 Conditional Data Space Characterization

Let us define the set of $\boldsymbol{x} \in \mathbb{R}^N$ which satisfies the conditions in (7) by

$$\mathcal{X} = \{\boldsymbol{x} \in \mathbb{R}^N \mid \mathcal{A}(\boldsymbol{x}) = \mathcal{A}(\boldsymbol{x}^{\mathrm{obs}}), \boldsymbol{q}(\boldsymbol{x}) = \boldsymbol{q}(\boldsymbol{x}^{\mathrm{obs}})\}.$$

Based on the second condition $\boldsymbol{q}(\boldsymbol{x}) = \boldsymbol{q}(\boldsymbol{x}^{\mathrm{obs}})$, the data in $\mathcal{X}$ is restricted to a line (see Sec 6 in [26], and [12]). Therefore, the set $\mathcal{X}$ can be re-written, using a scalar parameter $z \in \mathbb{R}$, as

$$\mathcal{X} = \{\boldsymbol{a} + \boldsymbol{b}z \mid z \in \mathcal{Z}\}, \text{ where } \mathcal{Z} = \{z \in \mathbb{R} \mid \mathcal{A}(\boldsymbol{a} + \boldsymbol{b}z) = \mathcal{A}(\boldsymbol{x}^{\mathrm{obs}})\}$$

with $\boldsymbol{a} = \boldsymbol{q}(\boldsymbol{x}^{\mathrm{obs}})$ and $\boldsymbol{b} = \Sigma \boldsymbol{\eta}_k (\boldsymbol{\eta}_k^\top \Sigma \boldsymbol{\eta}_k)^{-1}$. Now, let us denote a random variable $Z \in \mathbb{R}$ and its observation $z^{\mathrm{obs}} \in \mathbb{R}$, which satisfy $\boldsymbol{X} = \boldsymbol{a} + \boldsymbol{b}Z$ and $\boldsymbol{x}^{\mathrm{obs}} = \boldsymbol{a} + \boldsymbol{b}z^{\mathrm{obs}}$. Then, the selective $p$-value in (7) is re-written as

$$p_k^{\mathrm{selective}} = \mathbb{P}_{\mathrm{H}_{0,k}} \left( |\boldsymbol{\eta}_k^\top \boldsymbol{X}| > |\boldsymbol{\eta}_k^\top \boldsymbol{x}^{\mathrm{obs}}| \mid \boldsymbol{X} \in \mathcal{X} \right) = \mathbb{P}_{\mathrm{H}_{0,k}} \left( |Z| > |z^{\mathrm{obs}}| \mid Z \in \mathcal{Z} \right). \quad (8)$$

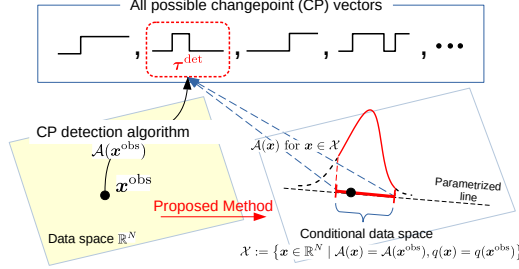

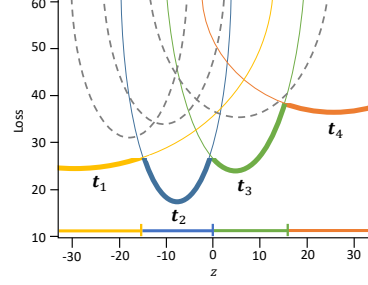

Figure 2: Schematic illustration of the proposed OptSeg-SI method. By applying a CP detection algorithm on the observed sequence $\boldsymbol{x}^{\mathrm{obs}}$, the optimal CP vector $\boldsymbol{\tau}^{\mathrm{det}}$ is obtained. In the OptSeg-SI method, the statistical inference is conducted conditional on the subspace $\mathcal{X}$ whose data has the same optimal CP vector as $\boldsymbol{x}^{\mathrm{obs}}$. We introduce a parametric programming method for efficiently characterizing the conditional data space $\mathcal{X}$.

Figure 3: A set of QFs each of which corresponds to a CP vector $\boldsymbol{\tau} \in \mathcal{T}_{k,n}$. The dotted grey QFs correspond to CP vectors that are not optimal for any $z \in \mathbb{R}$. A set $\{\boldsymbol{t}_1, \boldsymbol{t}_2, \boldsymbol{t}_3, \boldsymbol{t}_4\}$ contains CP vectors that are *optimal* for some $z \in \mathbb{R}$.

Since variable $Z \sim \mathbb{N}(0, \boldsymbol{\eta}_k^\top \Sigma \boldsymbol{\eta}_k)$ under the null hypothesis, the law of $Z \mid Z \in \mathcal{Z}$ follows a truncated Normal distribution. Once the truncation region $\mathcal{Z}$ is identified, the selective $p$-value in (8) can be computed as

$$p_k^{\mathrm{selective}} = F_{0, \boldsymbol{\eta}_k^\top \Sigma \boldsymbol{\eta}_k}^{\mathcal{Z}}(-|z^{\mathrm{obs}}|) + 1 - F_{0, \boldsymbol{\eta}_k^\top \Sigma \boldsymbol{\eta}_k}^{\mathcal{Z}}(|z^{\mathrm{obs}}|),$$

where $F_{m,s^2}^{\mathcal{E}}$ is the c.d.f. of the truncated Normal distribution with mean $m$, variance $s^2$ and the truncation region $\mathcal{E}$. Therefore, the main task is to identify $\mathcal{Z}$.

**Important notations.** In the rest of this paper, we use the following notations. Since we focus on a set of sequences parametrized by a scalar parameter $z \in \mathbb{R}$, we denote these sequences by

$$\boldsymbol{x}(z) = \boldsymbol{a} + \boldsymbol{b}z \tag{9}$$

or just simply by $z$. For a sequence with length $n \in [N]$, the set of all possible CP vectors with dimension $k \in [K]$ is written as $\mathcal{T}_{k,n}$. Given $\boldsymbol{x}(z)$, the loss of segmenting its first $n$ sub-sequence $\boldsymbol{x}(z)_{1:n}$ with a $k$-dimensional CP vector $\boldsymbol{\tau} \in \mathcal{T}_{k,n}$ is written as $L_{k,n}(z, \boldsymbol{\tau}) = \sum_{\kappa=1}^{k+1} C\left(\boldsymbol{x}(z)_{\tau_{\kappa-1}+1:\tau_\kappa}\right)$. For a subsequence $\boldsymbol{x}(z)_{1:n}$, the optimal loss and the optimal $k$-dimensional CP vector are respectively written as

$$L_{k,n}^{\mathrm{opt}}(z) = \min_{\boldsymbol{\tau} \in \mathcal{T}_{k,n}} L_{k,n}(z, \boldsymbol{\tau}), \quad \boldsymbol{T}_{k,n}^{\mathrm{opt}}(z) = \arg \min_{\boldsymbol{\tau} \in \mathcal{T}_{k,n}} L_{k,n}(z, \boldsymbol{\tau}). \tag{10}$$

Note that the notation $z \in \mathbb{R}$ in the definition (10) indicates that it corresponds to the sequence $\boldsymbol{x}(z)$.

**Main idea for identifying truncation region $\mathcal{Z}$.** Since we denoted $\boldsymbol{x}(z) = \boldsymbol{a} + \boldsymbol{b}z$ as in (9), truncation region $\mathcal{Z}$ is re-written as follows

$$\mathcal{Z} = \{z \in \mathbb{R} \mid \mathcal{A}(\boldsymbol{x}(z)) = \mathcal{A}(\boldsymbol{x}^{\mathrm{obs}})\} = \{z \in \mathbb{R} \mid \boldsymbol{T}_{K,N}^{\mathrm{opt}}(z) = \mathcal{A}(\boldsymbol{x}^{\mathrm{obs}})\}. \tag{11}$$

The main idea is to efficiently compute the optimal path of CP vectors $\boldsymbol{T}_{K,N}^{\mathrm{opt}}(z) \in \mathcal{T}_{K,N}$ for all values of $z \in \mathbb{R}$, which is computationally challenging. After $\boldsymbol{T}_{K,N}^{\mathrm{opt}}(z)$ is identified for all $z \in \mathbb{R}$, truncation region $\mathcal{Z}$ can be easily characterized, and the selective $p$-value in (8) can be computed.

### 3.2 Parametric CP detection

We introduce an efficient way to compute $\boldsymbol{T}_{K,N}^{\mathrm{opt}}(z)$ for all $z \in \mathbb{R}$. Although it seems intractable to solve this problem for infinitely many values of $z$, we can complete the task with a finite number of operations. Algorithm 1 shows the overview of our parametric CP detection method. Here, the algorithm is described in terms of general $n \in [N]$ and $k \in [K]$ along with a set of CP vectors $\hat{\mathcal{T}}_{k,n}$. In the current subsection, we set $n = N$, $k = K$ and $\hat{\mathcal{T}}_{k,n} = \mathcal{T}_{k,n}$. The case with general $n$, $k$ and $\hat{\mathcal{T}}_{k,n}$ will be discussed in §3.3.

**Algorithm 1** $\texttt{paraCP}(n, k, \hat{\mathcal{T}}_{k,n})$

---

**Input:** $n, k, \hat{\mathcal{T}}_{k,n}$

1: $u \leftarrow 1, z_1 \leftarrow -\infty, \boldsymbol{t}_1 \leftarrow \boldsymbol{T}_{k,n}^{\text{opt}}(z_u) = \arg\min\limits_{\boldsymbol{\tau} \in \hat{\mathcal{T}}_{k,n}} L_{k,n}(z_u, \boldsymbol{\tau})$

2: **while** $z_u < +\infty$ **do**

3:     Find the next breakpoint $z_{u+1} > z_u$ and the next optimal CP vector $\boldsymbol{t}_{u+1}$ such that
$$L_{k,n}(z_{u+1}, \boldsymbol{t}_u) = L_{k,n}(z_{u+1}, \boldsymbol{t}_{u+1}).$$

4:     $u \leftarrow u + 1$

5: **end while**

6: $U \leftarrow u$

**Output:** $\{(z_u, \boldsymbol{t}_u)\}_{u=1}^{U}$

---

In our parametric CP detection method, we exploit the fact that, for each CP vector $\boldsymbol{\tau} \in \mathcal{T}_{k,n}$, the loss function is written as a quadratic function (QF) of $z$ whose coefficients depend on $\boldsymbol{\tau} \in \mathcal{T}_{k,n}$. Since the number of possible CP vectors in $\mathcal{T}_{k,n}$ is finite, the parametric CP detection problem can be characterized by a finite number of these QFs. Figure 3 illustrates the set of QFs each of which corresponds to a CP vector $\boldsymbol{\tau} \in \mathcal{T}_{k,n}$. Since the minimum loss for each $z \in \mathbb{R}$ is the point-wise minimum of these QFs, the optimal loss function $L_{k,n}^{\text{opt}}(z)$ is the lower envelope of the set of QFs, which is represented as a piecewise QF of $z \in \mathbb{R}$. Parametric CP detection is interpreted as the problem of identifying this piecewise QF.

In Algorithm 1, multiple *breakpoints* $z_1 < z_2 < \ldots < z_U$ are computed one by one. Each breakpoint $z_u, u \in [U]$, indicates a point at which the optimal CP vector is replaced from one to the other in the piecewise QF. By finding all these breakpoints $\{z_u\}_{u=1}^{U}$ and the optimal CP vectors $\{\boldsymbol{t}_u\}_{u=1}^{U}$, the piecewise QF as in Figure 3 can be identified.

The algorithm is initialized at the optimal CP vector for $z = -\infty$, which can be easily identified based on the coefficients of the QFs. At step $u, u \in [U]$, the task is to find the next breakpoint $z_{u+1}$ and the next optimal CP vector $\boldsymbol{t}_{u+1}$. This task can be done by finding the smallest $z_{u+1}$ greater than $z_u$ among the intersections of the current QF $L_{k,n}(z, \boldsymbol{t}_u)$ and each of the other QFs $L_{k,n}(z, \boldsymbol{\tau})$ for $\boldsymbol{\tau} \in \mathcal{T}_{k,n} \setminus \{\boldsymbol{t}_u\}$. This step is repeated until we find the optimal CP vector when $z = +\infty$. The algorithm returns the sequences of breakpoints and optimal CP vectors $\{(z_u, \boldsymbol{t}_u)\}_{u=1}^{U}$. The entire path of optimal CP vectors for $z \in \mathbb{R}$ is given by $\boldsymbol{T}_{k,n}^{\text{opt}}(z) = \boldsymbol{t}_u, u \in [U]$, if $z \in [z_u, z_{u+1}]$.

### 3.3 Parametric DP

Unfortunately, parametric CP detection algorithm with the inputs $N$, $K$ and $\mathcal{T}_{K,N}$ in the previous subsection is computationally impractical because the number of all possible CP vectors $|\mathcal{T}_{K,N}|$ is exponentially increasing with $N$ and $K$. To resolve this computational issue, we utilize the concept of standard DP, and apply to parametric case, which we call *parametric DP*. The basic idea of parametric DP is to exclude the CP vectors $\boldsymbol{\tau} \in \mathcal{T}_{k,n}$ that cannot be optimal at any $z \in \mathbb{R}$.

**Standard DP (specific value of $z$).** In standard DP for a CP detection problem (for a specific $z$) with $N$ and $K$, we use $K \times N$ table whose $(k, n)^{\text{th}}$ element contains $\boldsymbol{T}_{k,n}^{\text{opt}}(z)$, the vector of optimal $k$ CPs for the subsequence $\boldsymbol{x}(z)_{1:n}$. The optimal CP vector for each of the subproblem with $n$ and $k$ can be used for efficiently computing the optimal CP vector for the original problem with $N$ and $K$.

Let $\texttt{concat}(\boldsymbol{v}, s)$ be the operator for concatenating a vector $\boldsymbol{v}$ and a scalar $s$. Then, it is known that the following equation, which is often called *Bellman equation*, holds:

$$\boldsymbol{T}_{k,n}^{\text{opt}}(z) = \arg\min_{\boldsymbol{\tau}(m)} \{L_{k,n}(z, \boldsymbol{\tau}(m))\}_{m=k}^{n-1}, \tag{12}$$

where $\boldsymbol{\tau}(m) = \texttt{concat}(\boldsymbol{T}_{k-1,m}^{\text{opt}}(z), m), m \in \{k, \ldots, n-1\}$. The Bellman equation (12) enables us to efficiently compute the optimal CP vector for the problem with $n$ and $k$ by using the optimal CP vectors of its sub-problems.

**Parametric DP (for all values of $z \in \mathbb{R}$).** Our basic idea is to similarly construct a $K \times N$ table whose $(k, n)^{\text{th}}$ element contains $\mathcal{T}_{k,n}^{\text{opt}} = \left\{ \boldsymbol{\tau} \in \mathcal{T}_{k,n} \mid \exists z \in \mathbb{R} \text{ s.t. } L_{k,n}^{\text{opt}}(z) = L_{k,n}(z, \boldsymbol{\tau}) \right\}$, which is

| **Algorithm 2** paraDP($\boldsymbol{x}(z)$, $K$) | **Algorithm 3** SI for Optimal CPs (OptSeg-SI) |
|---|---|
| **Input:** $\boldsymbol{x}(z)$ and $K$ | **Input:** $\boldsymbol{x}_{\mathrm{obs}}$ and $K$ |
| 1: **for** $k = 1$ to $K$ **do** | 1: $\boldsymbol{\tau}^{\mathrm{det}} \leftarrow \mathcal{A}(\boldsymbol{x}^{\mathrm{obs}})$ |
| 2:    **for** $n = 1$ to $N$ **do** | 2: **for** $\tau_k^{\mathrm{det}} \in \boldsymbol{\tau}^{\mathrm{det}}$ **do** |
| 3:       $\hat{\mathcal{T}}_{k,n} \leftarrow$ Lemma 1 | 3:    $\boldsymbol{x}(z) \leftarrow$ Eq.(9) |
| 4:       $\{(z_u, \boldsymbol{t}_u)\}_{u=1}^U \leftarrow$ paraCP($n, k, \hat{\mathcal{T}}_{k,n}$) | 4:    $\mathcal{T}_{K,N}^{\mathrm{opt}} \leftarrow$ paraDP($\boldsymbol{x}(z), K$) |
| 5:       $\mathcal{T}_{k,n}^{\mathrm{opt}} \leftarrow \{\boldsymbol{t}_u\}_{u=1}^U$ | 5:    $\mathcal{Z} \leftarrow \cup_{\boldsymbol{T}_{K,N}^{\mathrm{opt}}(z) \in \mathcal{T}_{K,N}^{\mathrm{opt}}} \{z : \boldsymbol{T}_{K,N}^{\mathrm{opt}}(z) = \mathcal{A}(\boldsymbol{x}^{\mathrm{obs}})\}$ |
| 6:    **end for** | 6:    $p_k^{\mathrm{selective}} \leftarrow$ Eq.(8) |
| 7: **end for** | 7: **end for** |
| **Output:** $\mathcal{T}_{K,N}^{\mathrm{opt}}$ | **Output:** $\{(\tau_k^{\mathrm{det}}, p_k^{\mathrm{selective}})\}_{k=1}^K$ |

a *set of CP vectors* that are optimal for some $z \in \mathbb{R}$. To identify $\mathcal{T}_{k,n}^{\mathrm{opt}}$, we construct a set $\hat{\mathcal{T}}_{k,n} \supseteq \mathcal{T}_{k,n}^{\mathrm{opt}}$, which is a set of CP vectors having potential to be optimal. In the same way as (12), we can consider Bellman equation for constructing $\hat{\mathcal{T}}_{k,n}$ as described in the following Lemma.

**Lemma 1.** *For $n \in [N]$ and $k \in [K]$, the set of CP vectors having potential to be optimal is constructed as $\hat{\mathcal{T}}_{k,n} = \cup_{m=k}^{n-1} \{\mathtt{concat}(\mathcal{T}_{k-1,m}^{\mathrm{opt}}, m)\}$, where we extend the* `concat` *operator for the case where the first argument is a set of vectors, which simply returns the set of concatenated vectors.*

In other words, the set $\hat{\mathcal{T}}_{k,n}$ can be generated from the optimal CP vectors of its sub-problems $\mathcal{T}_{k-1,m}^{\mathrm{opt}}$ for $m \in \{k, \ldots, n-1\}$. The proof for this result is deferred to Appendix A.1. From Lemma 1, we can efficiently construct $\hat{\mathcal{T}}_{k,n}$ which is subsequently used to identify $\mathcal{T}_{k,n}^{\mathrm{opt}}$. By repeating the recursive procedure and storing $\mathcal{T}_{k,n}^{\mathrm{opt}}$ in the $(k,n)^{\mathrm{th}}$ element of the table from smaller $n$ and $k$ to larger $n$ and $k$, we can end up with $\hat{\mathcal{T}}_{K,N} \supseteq \mathcal{T}_{K,N}^{\mathrm{opt}}$. By using parametric DP, the size of $\hat{\mathcal{T}}_{K,N}$ can be smaller than the size of all possible CP vectors $\mathcal{T}_{K,N}$, which makes the computational cost of paraCP($N, K, \hat{\mathcal{T}}_{K,N}$) substantially decreased compared to paraCP($N, K, \mathcal{T}_{K,N}$).

The parametric DP method is presented in Algorithm 2 and the entire OptSeg-SI method for computing selective $p$-values of the optimal CPs is summarized in Algorithm 3. Although they are not explicitly described in the algorithm, we also used several computational tricks for further reducing the size of $\hat{\mathcal{T}}_{k,n}$. See Appendix A.3 for the details.

## 4   Extension to Unknown $K$ Case

We present an approach for testing the significance of CPs detected by (2). The basic idea is the same as the proposed method for fixed $K$. With a slight abuse of notations, we use the following similar notations as the fixed $K$ case. For a sequence with length $n \in [N]$, the set of all possible CP vectors is written as $\mathcal{T}_n$. Given $\boldsymbol{x}(z)$ as in (9), the loss of segmenting its sub-sequence $\boldsymbol{x}(z)_{1:n}$ with a CP vector $\boldsymbol{\tau} \in \mathcal{T}_n$ is written as $L_n(z, \boldsymbol{\tau}) = \sum_{\kappa=1}^{\dim(\boldsymbol{\tau})+1} C\left(\boldsymbol{x}(z)_{\tau_{\kappa-1}+1:\tau_\kappa}\right) + \beta\dim(\boldsymbol{\tau})$. The optimal loss and the optimal CP vector on $\boldsymbol{x}(z)_{1:n}$ are respectively written as $L_n^{\mathrm{opt}}(z) = \min_{\boldsymbol{\tau} \in \mathcal{T}_n} L_n(z, \boldsymbol{\tau})$, $\boldsymbol{T}_n^{\mathrm{opt}}(z) = \arg\min_{\boldsymbol{\tau} \in \mathcal{T}_n} L_n(z, \boldsymbol{\tau})$[5].

**Identification of truncation region $\mathcal{Z}$.** To calculate $p_k^{\mathrm{selective}}$ for the $k^{\mathrm{th}}$ detected CP, we characterize the truncation region $\mathcal{Z} = \{z \in \mathbb{R} \mid \boldsymbol{T}_N^{\mathrm{opt}}(z) = \mathcal{A}(\boldsymbol{x}^{\mathrm{obs}})\}$, by computing $\boldsymbol{T}_N^{\mathrm{opt}}(z)$ for all $z \in \mathbb{R}$. We can slightly modify Algorithm 1 to the unknown $K$ case to compute $\boldsymbol{T}_N^{\mathrm{opt}}(z)$ for all $z \in \mathbb{R}$. Let $\mathcal{T}_n^{\mathrm{opt}}$ denote a set of CP vectors that are optimal at some $z \in \mathbb{R}$ for subsequence $\boldsymbol{x}(x)_{1:n}$ as $\mathcal{T}_n^{\mathrm{opt}} = \left\{\boldsymbol{\tau} \in \mathcal{T}_n \mid {}^{\exists}z \in \mathbb{R} \text{ s.t. } L_n^{\mathrm{opt}}(z) = L_n(z, \boldsymbol{\tau})\right\}$.

Since the set of all possible CP vectors $\mathcal{T}_N$ is huge, we use parametric DP with two additional computational tricks (Lemmas 2 and 3 below) for finding a substantially reduced set of CP vectors

$\hat{\mathcal{T}}_N \subseteq \mathcal{T}_N$ which contains all the optimal CP vectors for any $z \in \mathbb{R}$, i.e., $\hat{\mathcal{T}}_N \supseteq T_N^{\mathrm{opt}}$. The following two lemmas show how to construct $\hat{\mathcal{T}}_n$ by removing the CP vectors that never belong to $\mathcal{T}_n^{\mathrm{opt}}$.

**Lemma 2.** *For $m < n$, if a vector $\boldsymbol{\tau} \notin \mathcal{T}_m^{\mathrm{opt}}$, then* $\texttt{concat}(\boldsymbol{\tau}, m) \notin \mathcal{T}_n^{\mathrm{opt}}$.

**Lemma 3.** *For $m < n$, if $\boldsymbol{\tau} \notin \mathcal{T}_m^{\mathrm{opt}}$ and $L_m(z, \boldsymbol{\tau}) - \beta > L_m^{\mathrm{opt}}(z)$ for any $z \in \mathbb{R}$, then $\boldsymbol{\tau} \notin \mathcal{T}_n^{\mathrm{opt}}$.*

Proofs for these two lemmas are deferred to Appendix A.2. Based on Lemmas 2 and 3, $\hat{\mathcal{T}}_n$ can be constructed by $\hat{\mathcal{T}}_n = \cup_{m=k}^{n-1}\{\texttt{concat}(\mathcal{T}_m^{\mathrm{opt}}, m) \cup \mathcal{S}\}$, where $\mathcal{S}$ is a set of $\boldsymbol{\tau} \notin \mathcal{T}_m^{\mathrm{opt}}$ that does not satisfy Lemma 3. Then, we can use $\hat{\mathcal{T}}_n$ to find $\mathcal{T}_n^{\mathrm{opt}}$. We store $\mathcal{T}_n^{\mathrm{opt}}$ and continue this process recursively for larger $n$ until we get $\mathcal{T}_N^{\mathrm{opt}}$. After identifying $\mathcal{T}_N^{\mathrm{opt}}$, we can fully characterize truncation region $\mathcal{Z}$ and finally calculate selective $p$-values.

## 5   Numerical Experiments

We only highlight the main results. More details can be found in Appendix A.5.

**Methods for comparison.** We compared our OptSeg-SI method with SMUCE [14], which is an asymptotic test for multiple detected CPs, and SI for Binary Segmentation [18] (BinSeg-SI). It was reported that SI for Fused Lasso (proposed by the same authors), is worse than BinSeg-SI. Therefore, we only compared to BinSeg-SI. We additionally compared our method with SI method for optimal CPs with over-conditioning (OptSeg-SI-oc) to demonstrate the advantage of minimum conditioning. The details of OptSeg-SI-oc are shown in Appendix A.7 [6].

**Simulation setup.** Regarding false positive rate (FPR) experiments, we generated 1,000 null sequences $\boldsymbol{x} = (x_1, ..., x_N)$ in which $x_{i \in [N]} \sim \mathbb{N}(0, 1)$ for each $N \in \{10, 20, 30, 40\}$. In regard of testing the power, we generated sequences $\boldsymbol{x} = (x_1, ..., x_N)$ with sample size $N = 60$, in which

$$x_{i \in [N]} \sim \mathbb{N}(\mu_i, 1), \quad \mu_i = \begin{cases} 1 & \text{if } 1 \le i \le 20, \\ 1 + \Delta_\mu & \text{if } 21 \le i \le 40 \\ 1 + 2\Delta_\mu & \text{otherwise}, \end{cases}$$

for each $\Delta_\mu \in \{1, 2, 3, 4\}$. For each case, we ran 250 trials. Since the tests are performed only when a CP is selected, the power is defined as follows [18]:

$$\text{Power (or Conditional Power)} = \frac{\# \text{ correctly detected \& rejected}}{\# \text{ correctly detected}}.$$

A detection is considered to be correct if it is within $\pm 2$ of the true CP locations. Since it is often difficult to accurately identify exact CPs in the presence of noise, many existing CP detection studies consider a detection to be correct if it is within $L$ positions of the true CP locations [44]. We considered $L = 2$ to be consistent with our competitive method [18]. We used BIC [36] for the choice of $\beta$ when $K$ is unknown. We chose the significance level $\alpha = 0.05$. We used Bonferroni correction to account for the multiplicity in all the experiments.

**Experimental results.** Figures 4 and 5 respectively show the comparison results of the false positive rate (FPR) and true positive rate (TPR) when $K$ is fixed and $K$ is unknown. In both cases, since SMUCE guarantee is only asymptotic, it could not control the FPR when $N$ is small. While BinSeg-SI and OptSeg-SI-oc properly control the FPR, their powers are low because of over-conditioning. OptSeg-SI always has high power while properly controlling the FPR. Figure 6 shows the power demonstration of the OptSeg-SI method. While the existing methods missed many of true CPs, our method could identify almost all of them. Figure 7 shows the efficiency of OptSeg-SI method. We generated data for each case $(N, K) \in \{(200, 9), ..., (1200, 59)\}$. We ran 10 trials for each case.

Besides, we also conducted the following experiments to demonstrate the robustness of the OptSeg-SI method in terms of the FPR control:

• Non-normal data: we consider the data following Laplace distribution, skew normal distribution (skewness coefficient 10) and $t_{20}$ distribution. In each experiment, we generated 12,000 null sequences for $N \in \{10, 20, 30, 40\}$. We test the FPR for both $\alpha = 0.05$ and $\alpha = 0.1$. We confirmed that

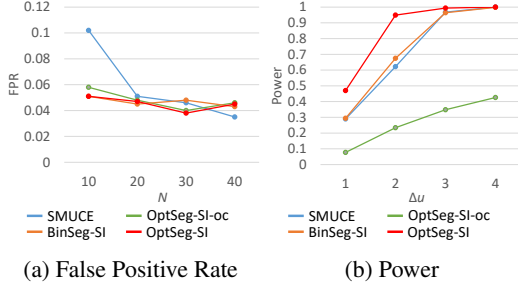

(a) False Positive Rate      (b) Power

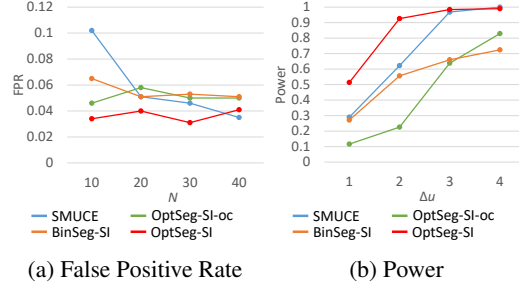

(a) False Positive Rate      (b) Power

Figure 4: False positive rate (FPR) and power comparison when $K$ is fixed.

Figure 5: False positive rate (FPR) and power comparison when $K$ is unknown.

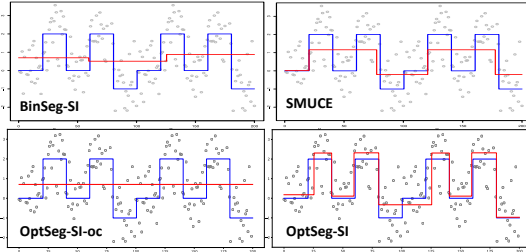

Figure 6: Power demonstration of the OptSeg-SI method. The underlying mechanism (blue), data points (grey), and the results of each method (red) are shown in each panel. The result of OptSeg-SI is mostly close to the ground truth compared to the other methods.

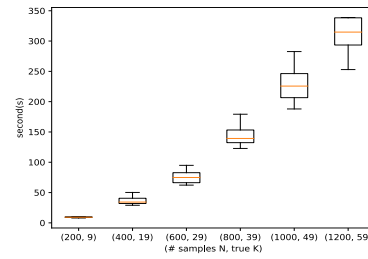

Figure 7: Computing time of the OptSeg-SI method. The computing time of our proposed method is almost linear.

Table 1: Power comparison on real-world bioinformatics related datasets.

|  | SMUCE | OptSeg-SI-oc | BinSeg-SI | OptSeg-SI |
|---|---|---|---|---|
| $\mathcal{D}_1$ | 0.53 | 0.24 | 0.33 | **0.75** |
| $\mathcal{D}_2$ | 0.62 | 0.27 | 0.32 | **0.71** |

our method still maintains good performance on FPR control. The results are shown in Appendix A.5. Besides, for dealing with the case of non-normal data, we can also apply a popular Box-Cox transformation [5] to the data before performing our method.

• Unknown $\sigma^2$: we consider the case when the variance is also estimated from the data. We generated 12,000 null sequences for $N \in \{50, 60, 70, 80\}$. Our OptSeg-SI method still can properly control the FPR. The results are shown in Appendix A.5.

We also performed TPR comparison on real-world dataset in which we showed that our method always has higher power compared to other existing method. We used *jointseg* package [35] to generate realistic DNA copy number profiles of cancer samples with "known" truth. Two datasets with 1,000 profiles of length $N = 60$ and true $K = 2$ for each were created as follows:

• $\mathcal{D}_1$: Resample from GSE11976 with tumor fraction $= 1$
• $\mathcal{D}_2$: Resample from GSE29172 with tumor fraction $= 1$

The results are shown in Table 1. Our proposed OptSeg-SI has higher power than the other methods in all cases. We also applied OptSeg-SI to the Array CGH data provided by Snijders et al. [37] and the Nile data which contains annual flow volume of the Nile river. All of the results are consistent with Snijders et al. [37], Jung et al. [21]. More details of the results can be found in Appendix A.6.

## 6 Conclusion

In this paper, we have introduced a powerful SI approach for the CP detection problem. We have conducted experiments on both synthetic and real-world datasets to show the good performance of the proposed OptSeg-SI method. In the future, we could extend our method to the case of multi-dimensional sequences [45]. For this case, computational efficiency is also a big challenge. Therefore, providing an efficient approach would also represent a valuable contribution.

## Broader Impact

Reliable machine learning (ML), which is the problem of assessing the reliability of data-driven knowledge obtained by ML algorithms, is one of the most important issues in the ML community. Changepoint (CP) detection is an important unsupervised learning task, and has been studied in many areas. Unfortunately, less attention has been paid to the statistical reliability of the detected CPs. Without statistical reliability, the results may contain many *false detections*. These falsely detected CPs are harmful when they are used for high-stake decision making.

The main idea of this paper is to employ a selective inference — a new promising approach for assessing the statistical reliability of data-driven hypotheses selected by complex data analysis algorithms — to quantify the reliability of the detected CPs. By mainly focusing on the reliability, this paper can have potential impact on reducing the risky as well as improving the quality of several CP detection-based data analysis tasks such as bioinformatics [14, 35], financial analysis [15], climatology [22], signal processing [19]. Especially for applications in healthcare domain, since the $p$-value that we introduced in the paper is valid and it is guaranteed that the probability of making false decisions is properly controlled, valid $p$-values can be used as one of many other possible criteria for making medical decisions.

## Acknowledgments and Disclosure of Funding

This work was partially supported by MEXT KAKENHI (20H00601, 16H06538), JST CREST (JPMJCR1502), RIKEN Center for Advanced Intelligence Project, and RIKEN Junior Research Associate Program.

## Footnotes

[3]The covariance matrix $\boldsymbol{\Sigma}$ is typically estimated by "null" sequences which are known to have no CP (see Takeuchi et al. [39] for an example in bioinformatics).

[4]In the unconditional case (6), the condition $\boldsymbol{q}(\boldsymbol{X}) = \boldsymbol{q}(\boldsymbol{x}^{\mathrm{obs}})$ does not change the sampling distribution since $\boldsymbol{\eta}_k^\top \boldsymbol{X}$ and $\boldsymbol{q}(\boldsymbol{X})$ are (marginally) independent. On the other hand, under the condition with $\mathcal{A}(\boldsymbol{X}) = \mathcal{A}(\boldsymbol{x}^{\mathrm{obs}})$, $\boldsymbol{\eta}_k^\top \boldsymbol{X}$ and $\boldsymbol{q}(\boldsymbol{X})$ are not conditionally independent. See Fithian et al. [12], Lee et al. [24] for the details.

[5]We recently noticed that $\ell_1$-penalty based SI for CP detection was extended to $\ell_0$-penalty [20], which results in a similar approach with the "unknown $K$ case" in our algorithm.

[6] We first developed OptSeg-SI-oc as our first SI method for optimal CPs detected by DP (unpublished). Later, its drawback (the over-conditioning) was removed by the OptSeg-SI method in this paper.

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
