[Supplementary Material]

# A    Appendix

## A.1    Proof for Lemma 1

**Lemma 1**. *For $n \in [N]$ and $k \in [K]$, the set of CP vectors having potential to be optimal is constructed as*

$$\hat{\mathcal{T}}_{k,n} = \cup_{m=k}^{n-1} \{\texttt{concat}(\mathcal{T}_{k-1,m}^{\text{opt}}, m)\}, \tag{13}$$

*where we extend the* `concat` *operator for the case where the first argument is a set of vectors, which simply returns the set of concatenated vectors.*

*Proof.* We prove the lemma by showing that any CP vector $\boldsymbol{\tau} \notin \mathcal{T}_{k-1,m}^{\text{opt}}$, for $m \in \{k, \ldots, n-1\}$, cannot be subvector of the optimal CP vectors for problems with larger $n$ and $k$ for any $z \in \mathbb{R}$, i.e., $\texttt{concat}(\boldsymbol{\tau}, m) \notin \mathcal{T}_{k,n}^{\text{opt}}$ for $n > m$. For $m \in \{k, \ldots, n-1\}$, let $\boldsymbol{\tau} \notin \mathcal{T}_{k-1,m}^{\text{opt}}$ be a CP vector which is NOT optimal for all $z \in \mathbb{R}$, i.e.,

$$L_{k-1,m}(z, \boldsymbol{\tau}) > L_{k-1,m}^{\text{opt}}(z) \quad \forall z \in \mathbb{R}.$$

It suggests that, for any $m \in \{k, \ldots, n-1\}$ and $z \in \mathbb{R}$,

$$
\begin{aligned}
L_{k,n}^{\text{opt}}(z) &= \min_{m' \in \{k,\ldots,n-1\}} \left( L_{k-1,m'}^{\text{opt}}(z) + C(\boldsymbol{x}(z)_{m'+1:n}) \right) \\
&\leq L_{k-1,m}^{\text{opt}}(z) + C(\boldsymbol{x}(z)_{m+1:n}) \\
&< L_{k-1,m}(z, \boldsymbol{\tau}) + C(\boldsymbol{x}(z)_{m+1:n})
\end{aligned}
$$

for all $z \in \mathbb{R}$. Thus, for any choice of $m \in \{k, \ldots, n-1\}$ and $z \in \mathbb{R}$, $\boldsymbol{\tau} \notin \mathcal{T}_{k-1,m}^{\text{opt}}$ cannot be a subvector of the optimal CP vector for problems with larger $n$ and $k$. In other words, only the CP vectors in $\cup_{m=k}^{n-1} \mathcal{T}_{k-1,m}^{\text{opt}}$ can be used as the subvector of optimal CP vectors for problems with larger $n$ and $k$.

## A.2    Proofs for Lemma 2 and 3 for the case when $K$ is unknown in §4

**Lemma 2**. *For $m < n$, if a vector $\boldsymbol{\tau} \notin \mathcal{T}_m^{\text{opt}}$, then $\texttt{concat}(\boldsymbol{\tau}, m) \notin \mathcal{T}_n^{\text{opt}}$.*

*Proof.* For $m < n$, if a vector $\boldsymbol{\tau} \notin \mathcal{T}_m^{\text{opt}}$,

$$L_m(z, \boldsymbol{\tau}) > L_m^{\text{opt}}(z) \quad \forall z \in \mathbb{R}.$$

It suggests that, for any $m \in \{0, \ldots, n-1\}$ and $z \in \mathbb{R}$,

$$
\begin{aligned}
L_n^{\text{opt}}(z) &= \min_{m' \in \{0,\ldots,n-1\}} \{L_{m'}^{\text{opt}}(z) + C(\boldsymbol{x}(z)_{m'+1:n}) + \beta\} \\
&\leq L_m^{\text{opt}}(z) + C(\boldsymbol{x}(z)_{m+1:n}) + \beta \\
&< L_m(z, \boldsymbol{\tau}) + C(\boldsymbol{x}(z)_{m+1:n}) + \beta.
\end{aligned}
$$

Therefore, for any $m \in \{0, \ldots, n-1\}$, if $\boldsymbol{\tau} \notin \mathcal{T}_m^{\text{opt}}$, then $\texttt{concat}(\boldsymbol{\tau}, m) \notin \mathcal{T}_n^{\text{opt}}$.

**Lemma 3**. *For $m < n$, if $\boldsymbol{\tau} \notin \mathcal{T}_m^{\text{opt}}$ and*

$$L_m(z, \boldsymbol{\tau}) - \beta > L_m^{\text{opt}}(z) \quad \forall z \in \mathbb{R}$$

*holds, then $\boldsymbol{\tau} \notin \mathcal{T}_n^{\text{opt}}$.*

*Proof.* For any $m \in \{0, \ldots, n-1\}$ and $z \in \mathbb{R}$, we have

$$
\begin{aligned}
L_n^{\text{opt}}(z) &= \min_{m' \in \{0,\ldots,n-1\}} \{L_{m'}^{\text{opt}}(z) + C(\boldsymbol{x}(z)_{m'+1:n}) + \beta\} \\
&\leq L_m^{\text{opt}}(z) + C(\boldsymbol{x}(z)_{m+1:n}) + \beta.
\end{aligned}
$$

For any $m \in \{0, \ldots, n-1\}$, if a CP vector $\boldsymbol{\tau} \notin \mathcal{T}_m^{\text{opt}}$ satisfies Lemma 3, then it suggests

$$
\begin{aligned}
& L_n^{\text{opt}}(z) \leq L_m^{\text{opt}}(z) + C(\boldsymbol{x}(z)_{m+1:n}) + \beta \\
\Leftrightarrow \quad & L_n^{\text{opt}}(z) < L_m(z, \boldsymbol{\tau}) - \beta + C(\boldsymbol{x}(z)_{m+1:n}) + \beta \\
\Leftrightarrow \quad & L_n^{\text{opt}}(z) < L_m(z, \boldsymbol{\tau}) + C(\boldsymbol{x}(z)_{m+1:n})
\end{aligned}
$$

for all $z \in \mathbb{R}$. On the other hand, we have

$$L_m(z, \boldsymbol{\tau}) + C(\boldsymbol{x}(z)_{m+1:n}) \leq L_n(z, \boldsymbol{\tau})$$

holds for any $z \in \mathbb{R}$ because the cost is always reduced when adding a changepoint at position $m$ without the penalty term. Hence, we have

$$L_n^{\mathrm{opt}}(z) < L_n(z, \boldsymbol{\tau})$$

for all $z \in \mathbb{R}$. Therefore, $\boldsymbol{\tau} \notin \mathcal{T}_n^{\mathrm{opt}}$ and Lemma 3 holds.

### A.3  Additional tricks for methods proposed in §3.

**Finding optimal CP vector when $z = -\infty$ in** $\mathtt{paraCP}(n, k, \hat{\mathcal{T}}_{k,n})$ **in Algorithm 1.** For each $\boldsymbol{\tau} \in \hat{\mathcal{T}}_{k,n}$, the corresponding loss function at $\boldsymbol{\tau}$ is written as a positive definite quadratic function. Therefore, at $z = -\infty$, the optimal CP vector is the one whose corresponding loss function $L_n(z, \boldsymbol{\tau})$ has the smallest coefficient of the quadratic term. If there are more than one quadratic function having the same smallest quadratic coefficient, we then choose the one that has the largest coefficient in the linear term. If those quadratic functions still have the same largest linear coefficient, we finally choose the one that has the smallest constant term.

**Additional pruning condition for parametric DP when $K$ is fixed.** In §3.3, we showed that $\mathcal{T}_{k,n}^{\mathrm{opt}}$ can be constructed from the set $\hat{\mathcal{T}}_{k,n} \subseteq \mathcal{T}_{k,n}$. By using the following lemma, we can construct a smaller superset of $\mathcal{T}_{k,n}^{\mathrm{opt}}$, which leads to further efficiency of parametric DP.

**Lemma 4.** *For $n \in [N]$, and $k \in [K]$, let*

$$\bar{\mathcal{T}}_{k,n} = \{\boldsymbol{\tau} \in \hat{\mathcal{T}}_{k,n-1} \setminus P_{\mathrm{prune}}\} \cup \{\mathtt{concat}(\mathcal{T}_{k-1,n-1}^{\mathrm{opt}}, n-1)\},$$

*where*

$$P_{\mathrm{prune}} = \{\boldsymbol{\tau} \in \hat{\mathcal{T}}_{k,n-1} \mid L_{k,n-1}(z, \boldsymbol{\tau}) > L_{k-1,n-1}^{\mathrm{opt}}(z), \forall z \in \mathbb{R}\}.$$

*Then $\mathcal{T}_{k,n}^{\mathrm{opt}} \subseteq \bar{\mathcal{T}}_{k,n} \subseteq \hat{\mathcal{T}}_{k,n}$.*

*Proof.* First, to show $\hat{\mathcal{T}}_{k,n} \supseteq \bar{\mathcal{T}}_{k,n}$, from (13),

$$\begin{aligned}
\hat{\mathcal{T}}_{k,n} &= \cup_{m=k}^{n-1} \{\mathtt{concat}(\mathcal{T}_{k-1,m}^{\mathrm{opt}}, m)\} \\
&= \cup_{m=k}^{n-2} \{\mathtt{concat}(\mathcal{T}_{k-1,m}^{\mathrm{opt}}, m)\} \cup \{\mathtt{concat}(\mathcal{T}_{k-1,n-1}^{\mathrm{opt}}, n-1)\} \\
&= \hat{\mathcal{T}}_{k,n-1} \cup \{\mathtt{concat}(\mathcal{T}_{k-1,n-1}^{\mathrm{opt}}, n-1)\} \\
&\supseteq \{\hat{\mathcal{T}}_{k,n-1} \setminus P_{\mathrm{prune}}\} \cup \{\mathtt{concat}(\mathcal{T}_{k-1,n-1}^{\mathrm{opt}}, n-1)\} = \bar{\mathcal{T}}_{k,n}.
\end{aligned}$$

Next, to show $\mathcal{T}_{k,n}^{\mathrm{opt}} \subseteq \bar{\mathcal{T}}_{k,n}$, we only need to prove that $\boldsymbol{\tau} \in P_{\mathrm{prune}}$ never be the optimal CP vector at $k, n$, i.e., $\boldsymbol{\tau} \notin \mathcal{T}_{k,n}^{\mathrm{opt}}$. For any $\boldsymbol{\tau} \in P_{\mathrm{prune}}$

$$\begin{aligned}
L_{k,n}(z, \boldsymbol{\tau}) &\geq L_{k,n-1}(z, \boldsymbol{\tau}) \\
&> L_{k-1,n-1}^{\mathrm{opt}}(z) \\
&= L_{k-1,n-1}^{\mathrm{opt}}(z) + C(x(z)_{n:n}) \\
&\geq \min_{m' \in \{k,\dots,n-1\}} (L_{k-1,m'}^{\mathrm{opt}}(z) + C(x(z)_{(m'+1):n})) \\
&= L_{k,n}^{\mathrm{opt}}(z),
\end{aligned}$$

for any $z \in \mathbb{R}$. Therefore, $\boldsymbol{\tau} \in P_{\mathrm{prune}}$ never belongs to $\mathcal{T}_{k,n}^{\mathrm{opt}}$. ∎

### A.4  Distribution of naive $p$-value and selective $p$-value when the null hypothesis is true

We demonstrate the *validity* of our proposed OptSeg-SI method by confirming the uniformity of $p$-value when the null hypothesis is true. We generated 12,000 null sequences $\boldsymbol{x} = (x_1, ..., x_N)$ in

which $x_{i \in [N]} \sim \mathbb{N}(0, 1)$ for each case $N \in \{10, 20, 30, 40\}$ and performed the experiments to check the distribution of naive $p$-values and selective $p$-values. From Figure 8, it is obvious that naive $p$-value does not follow uniform distribution. Therefore, it fails to control the false positive rate. The empirical distributions of selective $p$-value are shown in Figure 9. The results indicate our proposed method successfully control the false detection probability.

(a) $N = 10$       (b) $N = 20$       (c) $N = 30$       (d) $N = 40$

Figure 8: Distribution of naive $p$-value when the null hypothesis is true.

(a) $N = 10$       (b) $N = 20$       (c) $N = 30$       (d) $N = 40$

Figure 9: Distribution of selective $p$-value when the null hypothesis is true.

## A.5   Details for numerical experiments.

**Methods for Comparison.**   We compared the performance of the OptSeg-SI with the following approaches:

• **SMUCE [14].** This is asymptotic test for multiple detected CPs. The implementation of SMUCE is available at `https://cran.r-project.org/web/packages/stepR/index.html`.

• **[BinSeg-SI] SI for Binary Segmentation [18]** In Hyun et al. [18], it was reported that SI for Fused Lasso (proposed by the same authors), is worse than BinSeg-SI. Therefore, we only compare to BinSeg-SI. BinSeg-SI had been considered as a computationally efficient approximation of the problem in (7), where the authors additionally condition on extra information for computational tractability, e.g., the order that CPs are detected. This is one of the reasons why BinSeg-SI has low power. The implementation of BinSeg-SI is available at `https://github.com/robohyun66/binseginf`.

• **[OptSeg-SI-oc] SI method for optimal CPs with over-conditioning.** In SI, there are mainly two approaches to characterize the selection event. In the first approach, the selection event is only constructed based on the optimality condition of the problem, which is usually difficult or computationally impractical. Therefore, the second approach is used to overcome the computational challenge by additionally conditioning on extra event. Although the type I error can be properly controlled in the second approach, the power is generally low because of *over-conditioning*.

To see the advantage of minimum conditioning of the proposed method, we compare with two variants of SI for optimal CPs (each for fixed $K$ and unknown $K$ cases), which we call *OptSeg-SI-oc*. In each of these variants, instead of the truncation region $\mathcal{Z}$ characterized in the main paper, its subsets are used as the conditioning set. These subsets are constructed by considering all the operations when DP algorithm is used for detecting the optimal CPs. The OptSeg-SI-oc method and BinSeg-SI in Hyun et al. [18] are categorized as the second approach. We actually first developed OptSeg-SI-oc as our first SI method for optimal CPs (unpublished). The derivation of OptSeg-SI-oc is shown in Appendix A.7. Then, its drawback (over-conditioning) was resolved by the proposed OptSeg-SI method in this paper.

**p-value table**

| Location / Method | A | B | C | D | E |
|---|---|---|---|---|---|
| **BinSeg-SI** | 0 | $5 \times 10^{-14}$ | 0 | 1 | 0 |
| **OptSeg-SI-oc** | 0.004 | $4 \times 10^{-6}$ | 0.002 | 0.08 | 0.007 |
| **OptSeg-SI** | $6 \times 10^{-6}$ | $2 \times 10^{-12}$ | $4 \times 10^{-18}$ | $5 \times 10^{-41}$ | $7 \times 10^{-10}$ |

| Location / Method | F | G | H | I |
|---|---|---|---|---|
| **BinSeg-SI** | 0 | 1 | 0 | 1 |
| **OptSeg-SI-oc** | 0.7 | $4 \times 10^{-6}$ | 0.002 | 0.147 |
| **OptSeg-SI** | $6 \times 10^{-6}$ | $2 \times 10^{-12}$ | $4 \times 10^{-18}$ | $5 \times 10^{-41}$ |

Figure 10: Additional results for power demonstration. In the left figure, the blue line and the grey circles indicate the underlying mean and the observed sequence, respectively. The red dotted lines are the results of optimal segmentation (OptSeg) and binary segmentation (BinSeg) algorithms. Here, the CP detection results of OptSeg and BinSeg were the same. Then, the significance of each CP is tested. With Bonferroni correction, to control false detection rate at 0.05, the significance level is decided by $\frac{0.05}{9} \approx 0.006$. Three different $p$-values are shown for each detected CP: BinSeg-SI $p$-value, OptSeg-SI-oc $p$-value and OptSeg-SI $p$-value. BigSeg-SI missed many true CPs (**D**, **G**, **I**). This problem is the same for OptSeg-SI-oc (**D**, **E**, **F**, **I**). The OptSeg-SI method can identify all true CPs. The segments recovered based on the results of the significant testing from each method are shown in the right figure.

**Experimental Results.** We show the detail of experimental results as follows:

• **Additional experiment for power demonstration of the proposed method.** In Figure 10, we show additional results to demonstrate that our OptSeg-SI method can identify many true CPs.

• **The robustness of the proposed OptSeg-SI method in terms of the FPR control.**

  – Non-normal data: we considered the data following Laplace distribution, skew normal distribution (skewness coefficient 10) and $t_{20}$ distribution. In each experiment, we generated 12,000 null sequences for $N \in \{10, 20, 30, 40\}$. We tested the FPR for both $\alpha = 0.05$ and $\alpha = 0.1$. The FPR results are shown in Figure 11a, 11b and 11c. In case of Laplace distribution and skew normal distribution, our proposed method can properly control the FPR. For the case of $t_{20}$ distribution, the FPR is just a bit higher than the significance level.

  – Unknown $\sigma^2$: We generated 12,000 null sequences $\boldsymbol{x} = (x_1, ..., x_N)$, in which $x_{i \in [N]} \sim \mathbb{N}(0, 1)$, for $N \in \{50, 60, 70, 80\}$ and conducted experiments. In this case, the value of $\sigma^2$ is also estimated from the data. We first perform CP detection algorithm to detect the segments. Since the estimated variance tends to be smaller than the true value, we calculated the empirical variance of each segment and set the maximum value for $\sigma^2$. The results are shown in Figure 11d. Our proposed method still can properly control the FPR.

• **Comparison of FPR control when the sequence contains correlated data points.** In this experiment, we demonstrate that the asymptotic method (SMUCE) cannot control the FPR when the sequence contains correlated data points while our OptSeg-SI method can successfully control the FPR under the significance level $\alpha = 0.05$. We generated 1,200 null sequences $\boldsymbol{x} = (x_1, ..., x_N) \sim \mathbb{N}(\boldsymbol{\mu}, \boldsymbol{\Xi})$, where $N = 20$, $\boldsymbol{\mu} = (\mu_1, ..., \mu_N)$ in which $\mu_{i \in [N]} = 0$, and $\boldsymbol{\Xi} = \sigma^2 (\xi^{|i-j|})_{i,j \in [N]}$ in which $\xi$ is degree of correlation and $\sigma^2 = 1$. We conducted experiments for $\xi \in \{0.0, 0.2, 0.4, 0.6, 0.8\}$. The results are shown in Figure 12. When $\xi = 0.0$, i.e., there is no correlation between the data points, SMUCE can control the FPR at $\alpha = 0.05$. However, when $\xi$ increases, the FPR also increases. It indicates that SMUCE cannot control the FPR when the data points are correlated. On the other hand, our proposed OptSeg-SI method can successfully control the FPR under $\alpha$ in all cases.

(a) Laplace distribution

(b) Skew normal distribution

(c) $t_{20}$ distribution

(d) $\sigma^2$ is unknown

Figure 11: False positive rate of the proposed OptSeg-SI method when data is non-normal or $\sigma^2$ is unknown.

Figure 12: Comparison of FPR control when the sequence contains correlated data points. With SMUCE, the FPR increases when the degree of correlation increases. On the other hand, our proposed OptSeg-SI method can successfully control the FPR under $\alpha = 0.05$ in all cases.

## A.6 Details for real-data experiments.

**Array CGH data.** Array CGH analyses detect changes in expression levels across the genome. The dataset with ground truth was provided in Snijders et al. [37]. The results from our method were shown in Figure 13 and 14. The solid red line denotes the significant changepoint which has the $p$-value smaller than the significance level after Bonferroni correction. All of the results are consistent with Snijders et al. [37].

**Nile data.** The interest lies in unexpected event such as natural disasters. This data is the annual flow volume of the Nile river at Aswan from 1871 to 1970 (100 years). In Figure 15, the proposed algorithm results the changepoint at the $28^{\text{th}}$ position, corresponding to year 1899. This result is consistent with Jung et al. [21].

## A.7 Derivation of OptSeg-SI-oc mentioned in §5

As our first idea of SI for optimal CPs, we developed OptSeg-SI-oc. However, this method inherits the drawback of current SI studies (over-conditioning). Therefore, we have not officially published it yet. Later, we developed novel parametric programming techniques and proposed OptSeg-SI, which is presented in this paper, to address the over-conditioning problem. Here, we show the derivation of OptSeg-SI-oc.

(a) Chromosomes 1, 2, 3.

(b) Chromosomes 20, 21, 22.

Figure 13: Experimental results for cell line GM03576.

(a) Chromosome 14.

(b) Chromosomes 17, 18, 19

Figure 14: Experimental results for cell lines GM00143 and GM01750.

Figure 15: Experimental result for Nile data. The changepoint is detected at $28^{\text{th}}$ position which indicates there is a change in volume level in year 1899.

The main idea behind OptSeg-SI-oc is to characterize the conditional data space based on all steps of DP algorithm, i.e., performing inference conditional on all steps of DP. We focus on the case when $K$ is fixed, and it is easy to extend to the case when $K$ is unknown.

**Notation.** We denote $\mathcal{X}'$ as a conditional data space in OptSeg-SI-oc. The difference between $\mathcal{X}$ in §3.1 and $\mathcal{X}'$ here is that the latter is characterized with additional constraints on DP process. For an observed sequence $\boldsymbol{x}^{\text{obs}} \in \mathbb{R}^N$, its optimal CP vector is defined as $\boldsymbol{\tau}^{\text{det}}$. For a sequence with length $n \in [N]$, a set of all possible CP vectors with dimension $k \in [K]$ is defined as $\mathcal{T}_{k,n}$. Given

$\boldsymbol{x} \in \mathbb{R}^N$, the loss of segmenting its sub-sequence $\boldsymbol{x}_{1:n}$ with $\boldsymbol{\tau} \in \mathcal{T}_{k,n}$ is written as

$$L_{k,n}(\boldsymbol{x}, \boldsymbol{\tau}) = \sum_{\kappa=1}^{k+1} C(\boldsymbol{x}_{\tau_{\kappa-1}+1:\tau_\kappa}).$$

For a sub-sequence $\boldsymbol{x}_{1:n}$, the optimal loss and the optimal $k$-dimensional CP vector are respectively written as

$$L_{k,n}^{\mathrm{opt}}(\boldsymbol{x}) = \min_{\boldsymbol{\tau} \in \mathcal{T}_{k,n}} L_{k,n}(\boldsymbol{x}, \boldsymbol{\tau})$$

$$\boldsymbol{T}_{k,n}^{\mathrm{opt}}(\boldsymbol{x}) = \arg\min_{\boldsymbol{\tau} \in \mathcal{T}_{k,n}} L_{k,n}(\boldsymbol{x}, \boldsymbol{\tau}).$$

**Conditional data space characterization.** Since the inference is conducted conditional on all steps of DP, the conditional data space $\mathcal{X}'$ is written as

$$\mathcal{X}' = \left\{ \boldsymbol{x} \in \mathbb{R}^N \mid \bigcap_{k=1}^{K} \bigcap_{n=k}^{N} \boldsymbol{T}_{k,n}^{\mathrm{opt}}(\boldsymbol{x}) = \boldsymbol{T}_{k,n}^{\mathrm{opt}}(\boldsymbol{x}^{\mathrm{obs}}), q(\boldsymbol{x}) = q(\boldsymbol{x}^{\mathrm{obs}}) \right\}. \tag{14}$$

For simplicity, we denote $\boldsymbol{\tau}_{k,n}^{\mathrm{det}} = \boldsymbol{T}_{k,n}^{\mathrm{opt}}(\boldsymbol{x}^{\mathrm{obs}})$, the conditional data space $\mathcal{X}'$ can be re-written as

$$\mathcal{X}' = \left\{ \boldsymbol{x} \in \mathbb{R}^N \mid \bigcap_{k=1}^{K} \bigcap_{n=k}^{N} \boldsymbol{T}_{k,n}^{\mathrm{opt}}(\boldsymbol{x}) = \boldsymbol{\tau}_{k,n}^{\mathrm{det}}, q(\boldsymbol{x}) = q(\boldsymbol{x}^{\mathrm{obs}}) \right\}. \tag{15}$$

From the second condition, the data is restricted to the line [26, 12]. Therefore, the remaining task is to characterize the region in which $\boldsymbol{x} \in \mathbb{R}^N$ satisfies the first condition.

For each value of $k \in [K]$ and $n \in [N]$, $\boldsymbol{T}_{k,n}^{\mathrm{opt}}(\boldsymbol{x}) = \boldsymbol{\tau}_{k,n}^{\mathrm{det}}$ if and only if

$$\min_{\boldsymbol{\tau} \in \mathcal{T}_{k,n}} L_{k,n}(\boldsymbol{x}, \boldsymbol{\tau}) = L_{k,n}(\boldsymbol{x}^{\mathrm{obs}}, \boldsymbol{\tau}_{k,n}^{\mathrm{det}}) \tag{16}$$

$$\Leftrightarrow \qquad L_{k,n}^{\mathrm{opt}}(\boldsymbol{x}) = L_{k,n}(\boldsymbol{x}^{\mathrm{obs}}, \boldsymbol{\tau}_{k,n}^{\mathrm{det}}). \tag{17}$$

Based on the recursive structure of DP, we have

$$L_{k,n}^{\mathrm{opt}}(\boldsymbol{x}) = \min_{m \in \{k,\ldots,n-1\}} \left\{ L_{k-1,m}^{\mathrm{opt}}(\boldsymbol{x}) + C(\boldsymbol{x}_{m+1:n}) \right\}. \tag{18}$$

Combining (17) and (18), we have

$$L_{k-1,m}^{\mathrm{opt}}(\boldsymbol{x}) + C(\boldsymbol{x}_{m+1:n}) \geq L_{k,n}(\boldsymbol{x}^{\mathrm{obs}}, \boldsymbol{\tau}_{k,n}^{\mathrm{det}}), \tag{19}$$

for $m \in \{k,\ldots,n-1\}$. Since the cost function is in the quadratic form, (19) can be easily written in the form of $\boldsymbol{x}^\top A_{k,n,m} \boldsymbol{x} \leq 0$, where the matrix $A_{k,n,m} \in \mathbb{R}^{N \times N}$ depends on $k$, $n$ and $m$. It suggests that the conditional data space in (14) can be finally characterized as

$$\mathcal{X}' = \left\{ \boldsymbol{x} \in \mathbb{R}^N \mid \bigcap_{k=1}^{K} \bigcap_{n=k}^{N} \bigcap_{m=k}^{n-1} \boldsymbol{x}^\top A_{k,n,m} \boldsymbol{x} \leq 0, q(\boldsymbol{x}) = q(\boldsymbol{x}^{\mathrm{obs}}) \right\}.$$

Now that the conditional data space $\mathcal{X}'$ is identified, we can easily compute the truncation region and calculate $p$-value for each detected CP.