[Reviews · NeurIPS 2020]

Review 1

Summary and Contributions: The manuscript focuses on detecting change points and presents a novel method for computing p-values estimated from CPs detected through dynamic programming. The p-values are computed under a selective inference framework, producing valid p-values.

Strengths: The theoretical presentation is good and the DP algorithm is a computationally feasible algorithm as demonstrated in the empirical evaluation when applied to real world datasets. The algorithm is novel and solves a widely applicable problem. This work is very relevant to the NurIPS community due to the generality of the problem and practicality of the DP solution.

Weaknesses: The experiment section is very short, leaving all details to the appendix. Interpretation of the presented results is therefore very difficult. It would be great if some space was found for some minimal information about the experimental setup, which would increase clarity significantly. As it stands, the real world experiments are especially difficult to interpret.

Correctness: The claims are well explained and the method is solid. The empirical methodology is also good.

Clarity: The paper is easy to follow, however the empirical results are under detailed. A little bit more detail would go a long way to improving the clarity of the manuscript.

Relation to Prior Work: The manuscript is well placed within existing literature.

Reproducibility: Yes

Additional Feedback: This was a great read and a great method. Are you planning on releasing an implementation? Given the generality of the problem and the excellent results I'd imagine an implementation would be widely used.


Review 2

Summary and Contributions: Update after reading the author rebuttal: I'd like to thank the authors for their detailed rebuttal. I appreciate that they aim to expand the experimental section and related work, and simplify the notation and descriptions where possible. I certainly think these changes can make the paper even more accessible and hope that the authors will also incorporate the other suggestions made to improve the manuscript. With regards to the runtime shown in Figure 7, I am still not convinced this is "almost linear": connecting the mean of the boxplots does not seem to give a linear relationship, but more likely a quadratic one. I'd recommend the authors fit a polynomial to this data and work out whether it is linear or quadratic in the number of samples. The runtime of the method is still very reasonable, so it wouldn't be a problem if it is actually quadratic, but its possible to be scientific about this and simply report, for instance, "we find the runtime complexity in practice is O(N^alpha K^beta)" for some values of alpha and beta. It would also be good if the authors can clarify in the manuscript that the pruning steps do not remove the theoretical exponential runtime complexity. I appreciate that the authors have responded to my comments on the broader impact statement and hope they will incorporate the nuance they provide in their rebuttal in the revised manuscript. --- Change point detection algorithms generally return a set of detected change points. This paper describes a method for estimating the p-values associated with the change points detected through optimal partitioning. The main challenge in this problem is that using the same data for detecting change points and computing p-values leads to biased results. To overcome this the authors present a theoretical treatment of selective inference applied to optimal partitioning, which allows for exactly estimating the p-values associated with the detected change points. The resulting algorithm is evaluated on synthetic and real datasets.

Strengths: This work presents an extensive and thorough treatment of the problem of estimating accurate p-values for change points found by optimal partitioning. While many methods for change point detection exists, to the best of my knowledge few have considered estimating the p-values of the detected change points. This is relevant in practice as it allows to evaluate the significance of a detected change point. Moreover the theoretical work could be relevant to the broader ML community.

Weaknesses: In comparison to the depth of the theoretical treatment, the numerical experiments are rather brief. Without the additional details in the supplementary material it is difficult to evaluate the numerical results fully. While I understand the space limitation, the experimental section could have been given more attention. For instance, Table 1 shows the results for two "bioinformatics related datasets", but no context is provided about their source, descriptive statistics, or true number of change points. Thus in its current form Table 1 simply shows numbers that are larger for the proposed method, which in itself isn't very informative.

Correctness: Overall the presented methodology appears to be correct, although I have not checked all proofs in full detail. With regards to Algorithm 1, the authors mention in Sec. 3.3 that the number of possible CP vectors increases exponentially. Subsequently a reduced set of possible CP vectors is introduced, which certainly would reduce the runtime of Algorithm 1. However, unless I've missed it, it is not quantified *how much* smaller this set is. Does the complexity of Algorithm 1 still grow exponentially but at a smaller rate? A relatively minor note is that the authors write in the caption of Figure 7 that the runtime is almost linear. Judging by the means of the boxplots however it seems that the runtime is more likely to be quadratic. It is also not clear if this runtime includes the change point detection step as well, or whether it is only the runtime for computing the p-values.

Clarity: The paper is overall well written and clearly structured. However, the theoretical description is fairly dense, with quite a lot of different notation that needs to be introduced to present the method. This can make the description occasionally difficult to follow. I believe some of the notation can be clarified or streamlined to alleviate this problem. For example, while change point configurations are generally described with lowercase bold $\tau$, eq (9) introduces uppercase bold T^{opt} as a change point configuration (instead of bold lowercase $\tau^{opt}$). Similarly, Sec. 3.2 introduces lowercase bold $t$ for change point vectors, instead of bold $\tau$. This might lead to unnecessary confusion. Another approach that may help clarify the many different quantities that are introduced is to provide a description in language. For instance, (6) could be described as "the probability that the absolute difference in mean for a random sequence is equal or greater to that of the observed sequence, given the same set of change points, ..." It would be good if the authors mention early on that their method is for univariate time series. While this fact can be extracted from the definition of x^{obs}, it would be good to clarify.

Relation to Prior Work: The related work on estimating p-values in change point detection is discussed properly, as is the related work on selective inference. With regards to the dynamic programming algorithm and the computational tricks in Lemma's 1-3 to speed it up, it seems that this paper might not be the first to introduce these for change point detection (see [1-3], among others). It would be good if this is clarified in case these overlap with those in earlier works. One stream of related work that is not discussed in this manuscript is that on Bayesian online change point detection (i.e. [4, 5] and subsequent works). It could be argued that existing Bayesian methods already give posterior probabilities for the existence of a change point at every time step. While I appreciate that this work takes a hypothesis-testing approach, it seems that it would be appropriate to discuss the differences between the two streams of research. [1]: Rigaill - Pruned dynamic programming for optimal multiple change-point detection, 2010. [2]: Killick et al - Optimal detection of changepoints with a linear computation cost (2011) [3]: Maidstone et al. - On optimal multiple changepoint algorithms for large data (2017) [4]: Adams & Mackay - Bayesian Online Change Point Detection (2007) [5]: Fearnhead & Liu - On-line inference for multiple changepoint problems (2007)

Reproducibility: Yes

Additional Feedback: Minor comments and questions: - The caption of Figure 1 states that BinSeg p-values fail to detect some true CPs (C, D) due to the lack of power. Shouldn't this be "(C, D, F)"? - In Figure 4 & 5, $\Delta u$ is not explained, and the appendix is needed to make sense of the results in these figures. - The authors may want to consider referring the reader to some review papers on change point detection for background information, such as Aminikhanghahi & Cook (2017), Truong et al (2020), and Van den Burg & Williams (2020). Extremely minor comments: - In (5) and several other equations and definitions $\Sigma$ is not bold, whereas it is bold when introduced. - Footnotes typically follow punctuation - Caption Figure 2: programing -> programming - Line 100: "of a test statistic" - Line 173: fix "and apply to parametric case" - Line 174: "which" -> "that" - Line 198: "which makes ... can be substantially decreased" (unclear sentence) - Line 241: "almost *all* of them" Comments on broader impact statement: The authors give no discussion of potential negative impact of their work. They do note that the p-value is useful in the healthcare domain "for making medical decisions". Personally, I would be quite concerned if my physician made medical decisions based on p-values, and considering the many reports on the replication crisis and the role of p-values therein, this may not be the best example. In fact, it may actually be better suited as an example of the potentially negative effects of using the p-values obtained through this work.


Review 3

Summary and Contributions: The paper proposes a method for more reliable statistical evaluation of change point detection, by computing an exact p-value (and hence allowing to search for the best change point vector). The exact p-value is based on the use of 'selective inference'. The authors propose a variant that does not result in over-conditioning, and to combat the computational effort require they propose a dynamic programming based approach. The resulting method is evaluated on artificial and real data and shown to outperform other CP detection methods, including an earlier CP method based on selective inference.

Strengths: + theoretically well founded + well structured and written paper with a mostly clear explanation, though dense at times + clever dynamic programming variant for computing the optimal CP vectors for a z + experiments demonstrate the meaningful results obtained, compared to state of the art methods

Weaknesses: - dense writing - computation time is reported, but only on itself, not in comparison to the other methods

Correctness: yes, having checked as much of the main paper as I could

Clarity: yes, well structured and written

Relation to Prior Work: yes, good discussion of related work and comparison

Reproducibility: Yes

Additional Feedback:


Review 4

Summary and Contributions: The authors introduce a selective inference approach to assess the p-value of a detected changepoint. The p-value is conditioned on the number of random line segments having the same optimal CP locations. which is made possible by incorporating several computational tricks.

Strengths: The paper is very well written and extremely accessible. The method is novel and clearly described. The results on simulated and real world data are convincing. The broader impact is given since the proposed methodology advances the field by increasing the statistical reliability of detected changepoints.

Weaknesses: One very minor comment is the low number of real world data sets used to validate the method. I am very aware of the problem that real world data sets to validate changepoint detection approaches are hard to find. I found the problem to be that usually, the location and existence of ground truth changepoints are debatable. Allow me to share a selection of some data sets that I have found to be useful in the past (sometimes appropriate transformations have to be applied, e.g. log-diff): Transatlantic internet traffic has daily and weekly changes in mean (https://ieeexplore.ieee.org/abstract/document/1716452 2006) Crest toothpaste market share increases due to a change in management and marketing (Section 1.1.4 https://onlinelibrary.wiley.com/doi/pdf/10.1002/9781118032978#page=281) The seasonally adjusted monthly US civilian unemployment rate changes due to business cycles (Montgomery, Forecasting the US unemployment rate https://www.tandfonline.com/doi/abs/10.1080/01621459.1998.10473696) The public drunkenness intakes in Minneapolis has two ground truth changepoints when effective political (counter-)interventions took place (Meidinger, Applied Time Series Analysis for the Social Sciences, 1980) The yearly global average temperature changes due to climate change (https://data.giss.nasa.gov/gistemp/) I also suggest to add one more internal comparison partner to the experiments that derives the significance of changepoints with a Bonferroni correction `OptSeg-BF`, as in Figure 1 (naive). Although correcting by the number of detected changepoints is reasonable, the authors mention the problem of double-dipping with this approach. Therefore, an equally justified way of correcting would incorporate that the segmentation method already searched a larger space of hypothesis before returning the changepoint candidates. A stricter Bonferroni correction can therefore be justified, e.g. by dividing by the all possible configurations of changepoint locations (alpha/(n choose k)) or by the number of possible change point locations (alpha/n). Doing that, the Bonfferroni correction would be a bit more conservative. In Figure 1, 0.05/1192052400 = 4e-11 or 0.05/100 = 0.0005 -- the second approach giving the desired results. Adding on that, maybe you could briefly justify why double-dipping is not an issue for the unknown K case, where you return segmentations for all K where the p-value for each changepoint in the kth case is unaffected by the other cases of k.

Correctness: Yes

Clarity: Yes

Relation to Prior Work: Yes

Reproducibility: Yes

Additional Feedback:

[Author Response · NeurIPS 2020]

We would like to thank the reviewers for their constructive feedback. We will correct the misprints raised and include the suggestions for improving the readability of the paper accordingly.

# 1 Reviewer #1

**The experiment section is short.** We will optimize the space and move some information about the experimental setup from Appendices A.5 and A.6 to Section 5 of the main paper.

**Releasing an implementation.** We will definitely release the code that was provided in the supplementary materials accompanying this submission.

# 2 Reviewer #2

**The experiments are brief.** We will add more information about the experimental setup as well as the interpretation of the results from Appendices A.5 and A.6 to Section 5 of the main paper.

**Worst-case complexity of Algorithm 1 and runtime in Figure 7.** In the worst-case, the complexity of Algorithm 1 still grows exponentially. This is a common issue in other parametric programming applications such as regularization paths. However, fortunately, it has been well-recognized that this worst case rarely happens in practice. In Figure 7, the runtime is almost linear thanks to the efficiency of the pruning lemmas. However, in the worst-case—which rarely happens in practice—our algorithm still has quadratic complexity.

**Notations are complicated.** Thanks for the suggestion to simplify the notation as well as the recommendation of providing description in language. We will do our best to improve the readability of the paper.

**Relation to prior work.** In several previous studies, similar arguments to Lemmas 1-3 have been made for a single specific value of $z$. However, the main difference in our work is that our Lemmas 1-3 are for all $z \in \mathbb{R}$, which is technically much more challenging and requires parametric programming techniques to conduct the pruning efficiently. Regarding the related works on Bayesian change point detection, we thank the reviewer for the suggestions. We will provide some appropriate discussions in the related work part in Section 1 of the paper.

**Additional suggestions from the reviewer.** Regarding the caption of Figure 1, as the reviewer pointed out, it should be "(C, D, F)" instead of "(C, D)". In regards to the explanation of $\Delta\mu$ in Figure 4 and 5, we will move the explanation from Appendix A.5 to the main paper. In terms of referring the reader to some review papers, we have already cited Truong et al (2019). We will also refer to Aminikhanghahi & Cook (2017) and Van den Burg & Williams (2020).

**Broader impact statement.** We are not saying that we solely rely on $p$-values for making medical decisions. With the valid $p$-values that we introduced in the paper, it is guaranteed that the probability of making false decisions is properly controlled, and valid $p$-values can be used as one of many other possible criteria for making medical decisions.

# 3 Reviewer #3

**Dense writing.** We will simplify the writting as much as possible.

**Computation time is reported, but only on itself, not in comparison to other methods.** The baseline methods are faster than the proposed method, but they have much lower power than the proposed method. Besides, SMUCE—one of the baseline methods—is only asymptotic and can not theoretically guarantee to properly control the false positive rate (FPR). The reason why we include the computational time of the proposed method in the paper is that we want to demonstrate our method not only has higher power but also is practically useful. However, based on the suggestion of the reviewer, we will also consider providing the computational time of the baseline methods in Section 5 or in Appendix A.5 of the revised version.

# 4 Reviewer #4

**Real world datasets with ground-truth.** Thanks for sharing with us a selection of real world datasets with ground-truth. We will investigate and apply our method to them.

**Suggestion of adding one more internal comparison partner.** Thanks for the suggestion. We will clearly explain how we can correct the multiplicity from the selective $p$-values. Actually, we have already considered Bonferroni correction in Figure 1. Once the selective $p$-values are obtained, we can treat these as nominal $p$-values, and any multiple testing adjustment such as Bonferroni correction can be applied as long as they are nominal $p$-values [4].

[Meta-Review · NeurIPS 2020]

This paper describes a method for estimating the p-values associated with the change points detected through optimal partitioning. The reviewers were unanimous in their vote to accept. Someone stated that it was a "great read and a great method" and wants to use their code.